# Theoretical and Experimental Studies of the Impact of High-Speed Raindrops on the Structural Elements of Modern Technology

**DOI:** 10.3390/ma15207305

**Published:** 2022-10-19

**Authors:** Ying Sun, Polina F. Pronina, Lev N. Rabinskiy, Olga V. Tushavina

**Affiliations:** 1Department of Mechanical Engineering, Hangzhou Xiaoshan Technician College, 311200 Tonghui South Road 448, Xiaoshan District, Hangzhou 311231, China; 2Department of Management of the Operation of Rocket and Space Systems, Moscow Aviation Institute (National Research University), Volokolamskoe Highway 4, 125993 Moscow, Russia; 3Department of Strength of Materials, Dynamics and Strength of Machines, Moscow Aviation Institute (National Research University), Volokolamskoe Highway 4, 125993 Moscow, Russia

**Keywords:** single-jet generator, water droplet impact, round plate, Fourier method, influence functions

## Abstract

Objects of modern technology located in the rain zone experience additional loads and can be destroyed due to water droplet erosion. With a significant number of successive impacts in a certain period of time, rain causes damage to the surface of materials or structures. It should be noted that supersonic water droplet impact has a low probability of occurrence; however, the peak pressure impulse of the water hammer (up to GPa level) far exceeds the strength of many materials, and a small number of impacts are enough to damage the material or structure. Therefore, it is very interesting to determine the external load caused by a water droplet’s impact and its response to various obstacles. In this work, the external load is determined on the basis of experimental studies. To carry out such tests, a single-jet generator is most widely used, which, with a certain ease of operation, makes it possible to investigate the mechanisms of damage to materials and the effect of water droplet impact erosion on structural elements. Based on the obtained research results, mathematical modeling of the droplet impact with an obstacle is provided. The examples are considered.

## 1. Introduction

Objects of modern technology move at a high speed in the rain zone, and so perceive additional loads in the form of water droplet impact erosion and can be destroyed.

Due to their relatively high speed, water droplet impacts on modern structural elements cannot be underestimated. Such high-velocity water droplet impact easily causes the erosion of windshields, antenna radomes, infrared windows and engine nacelles [1,2,3,4,5]. Especially for military aircraft, the problem of rain erosion is more serious, and the surface of the material will be eroded or even chipped [6]. Therefore, it is very important to evaluate the corrosion resistance of aircraft parts that are easily affected by water droplet impacts [7].

Typically, water droplet impact failure problems occur with windshield fairings, which are constructed from a high-molecular weight polymer composite material. Due to this material’s excellent optical properties, high mechanical strength, low specific gravity, low thermal conductivity and ease of processing and molding, it is widely used in windshields, cockpit lights and aircraft windows [8].

In the past few decades, many scientists have studied the damage to aircraft plexiglass under high-speed water jet conditions [7,9]. Bowden and Brunton [10] summarized the typical forms of surface failure of polymers such as polymethyl methacrylate, including a central undamaged area surrounded by annular depressions or cracks and short annular cracks. In addition, high-velocity fluid impact also causes stress waves to propagate within the material. Internal damage caused by the interaction of stress waves is sometimes more severe than surface damage. In the case of a limited thickness of the material, water droplet impact causes damage to the rear surface and the central zone due to the stress wave [11].

The study of hydraulic shock pressure is at the theoretical stage due to the complexity of modeling the process.

In this work, in addition to the theoretical stage, mathematical modeling of the processes of the water droplet impact with an obstacle is performed.

The non-stationary interaction of an inhomogeneous round plate with an external load simulating high-velocity rain pressure is considered for the first time. A mathematical model is constructed, on the basis of which the kinematic and dynamic parameters of the system are obtained. Based on the obtained results, it is possible to predict the stress–strain state of aircraft structural elements during a flight through the rain zone at a high speed.

## 2. Theoretical Basis

To analyze the mechanism of internal damage, many scientists use the photoelasticity of plexiglass to observe the propagation of an internal stress wave and the damage process after a water droplet impact [12]. Bowden et al. [13] also explained stress wave interactions through theoretical derivation. In addition, the water hammer pressure caused by a liquid–solid impact is also difficult to study. Due to the complexity of the impact process, the study of water hammer pressure is at a theoretical stage.

In this paper, impact testing of composite plexiglass for aviation was carried out using high-velocity jets generated by a single-jet device and compared their damage characteristics. At the same time, the sample damage mechanism was analyzed by observing the internal stress wave.

A mathematical model of the water droplet impact interaction of rain pressure with an inhomogeneous round plate was constructed.

## 3. Methodology. Conducting Experimental Studies

The test was carried out on a platform based on a 10 mm caliber light gas gun, and the layout is shown in Figure 1. The working principle of the platform was as follows: using high-pressure gas to feed a bullet into a stainless steel nozzle that stores water and creating high-speed jets by compressing the water in the nozzle. Before testing, a small amount of clean water was pre-injected into the nozzle and hermetically sealed with neoprene. The speed of the bullet was controlled by controlling the pressure in the gas chamber, which could create a jet at a speed from 90 m/s to 700 m/s.

To monitor the shape of the jet in real time and calculate the jet velocity, the impact process was observed using a high-speed camera.

Figure 2 shows the law of changing the shape and speed of the jet, and it can be seen that the speed of the jet gradually increases after injection, the diameter of the jet gradually increases, and air resistance acts on the jet head, giving it a stable spherical shape. Because the shape of the jet head is similar to that of a raindrop, the jet could be used to simulate real high-speed droplet impact. The jet velocity and head shape reached a steady state about 10 mm from the nozzle opening, and the jet gradually dissipated as the distance increased.

In this work, this platform was used to investigate the damage behavior of oriented and non-oriented plexiglass for aviation when exposed to a high-speed jet. A nozzle with a diameter of 0.8 mm was used in the tests, and the resulting jet had a diameter of about 4.5 mm. The presented results of experimental studies allow us to conclude that at a sufficient distance from the nozzle, the jet velocity has a steady character (Figure 2). Based on this, in the first approximation, it is possible to consider the external load acting on the barrier as suddenly applied, which makes it possible to use a mathematical model of the interaction of round plates under non-stationary impact.

## 4. Results and Discussion

### 4.1. Typical Form of Damage and Characteristics of Specimens after Impact

Figure 3 shows microscopic images of the surface of two plexiglass samples after a jet impact at a speed of 320 m/s, and the difference between the forms of damage on the two types of samples is well expressed. It shows that damage to non-oriented plexiglass is mainly concentrated on the surface, including an almost circular central undamaged area (1), an annular recessed area (2) and a peripheral short annular crack (3). It should be noted that there are no annular short cracks caused by high-velocity lateral jet erosion beyond the periphery of the intact area. At the same time, a typical river pattern appeared in brittle materials in the destruction zone of the annular microcrack, and a number of steps were located in the direction of crack propagation on the surface of the river pattern and gradually merged during expansion, forming more obvious steps at the far end of the damaged area.

### 4.2. Mathematical Modeling of the Interaction of Round Plates with Non-Stationary Droplet Impact

We consider axisymmetric vibrations of a round plate under the action of a non-stationary droplet impact. The plate appears to be isotropic with average characteristics. Droplet impact is modeled by external non-stationary pressure p, directed perpendicular to the plate surface (Figure 4).

The raindrops are assumed to be spherical and uniform in size and do not deform.

The initial-boundary value problem for a circular axisymmetric plate is as follows [14]:

The equation of motion in displacements is:(1)$$\begin{array}{r} \gamma^{2}\frac{\partial^{2}w}{\partial \tau^{2}}=-\Delta^{2}w+P,\;\\ \Delta =\frac{1}{r}\frac{\partial }{\partial r}\left(r\frac{\partial }{\partial r}\right)=\frac{\partial^{2}}{\partial r^{2}}+\frac{1}{r}\frac{\partial }{\partial r}. \end{array}$$

The initial conditions are assumed to be homogeneous:(2)$${\left.w\right|}_{\tau =0}=0,{\left.\;\frac{\partial w}{\partial \tau }\right|}_{\tau =0}=0.$$

The boundary conditions also have a homogeneous form:(3)$${\left.w\right|}_{r=1}=0,{\left.M\right|}_{r=1}={\left.-\left(\frac{\partial^{2}w}{\partial r^{2}}+\frac{ℵ}{r}\frac{\partial w}{\partial r}\right)\right|}_{r=1}=0.$$

In addition to the boundary conditions (3), it is necessary to set the conditions for the limited deflection of the plate w and changes in curvature, which are the conditions for the continuity of the median plane.
(4)$$w=O\left(1\right),\;\frac{\partial w}{\partial r}=O\left(r\right),r\to 0.$$

The solution of the problem can be represented in the integral form: (5)$$\begin{array}{l} w\left(r,\tau \right)=\int_{0}^{1}G_{w}\left(r,\tau ;\xi \right)*p\left(\xi ,\tau \right)d\xi +\\ +\gamma^{2}\left[\int_{0}^{1}\frac{\partial G_{w}\left(r,\tau ;\xi \right)}{\partial \tau }\varphi \left(\xi \right)d\xi +\int_{0}^{1}G_{w}\left(r,\tau ;\xi \right)\psi \left(\xi \right)d\xi \right]. \end{array}$$
where w,τ,ℵ,r are the dimensionless deflection, time, curvature and radius of the plate, respectively, $G_{w}\left(r,\tau ;\xi \right)$ is the Green’s function for deflection, $p\left(\xi ,\tau \right)$ is the external load caused by drop impact, and ∗ is the convolution operation. The Green’s function for the deflection is determined from the following initial-boundary value problem:(6)$$\gamma^{2}\frac{\partial^{2}G_{w}}{\partial \tau^{2}}=-\Delta^{2}G_{w}+\delta \left(\tau \right)\delta \left(r-\xi \right);$$
(7)$$\begin{array}{c} {\left.G_{w}\left(r,\tau \right)\right|}_{\tau =0}=0,{\left.\frac{\partial G_{w}\left(r,\tau \right)}{\partial \tau }\right|}_{\tau =0}=0;\\ {\left.G_{w}\left(r,\tau \right)\right|}_{r=1}=0,\\ {\left.\left(\frac{\partial^{2}G_{w}\left(r,\tau \right)}{\partial r^{2}}+\frac{ℵ}{r}\frac{\partial G_{w}\left(r,\tau \right)}{\partial r}\right)\right|}_{r=1}=0. \end{array}$$
(8)$$\begin{array}{l} G_{w}\left(r,\tau \right)=O\left(1\right),\;\frac{\partial G_{w}\left(r,\tau \right)}{\partial r}=O\left(r\right),r\to 0. \end{array}$$
where $\delta \left(x\right)$ is the Dirac delta function.

The angle of rotation ϑ, moment M and shear force Q are also determined by Equation (5), in which $G_{w}\left(r,\tau ;\xi \right)$ must be replaced by the corresponding influence functions $G_{\vartheta }\left(r,\tau ;\xi \right),\;G_{M}\left(r,\tau ;\xi \right)$ and $G_{Q}\left(r,\tau ;\xi \right)$, respectively. The latter are calculated according to known formulas [15,16]. For determining $G_{w}\left(r,\tau ;\xi \right)$, we use the Fourier method, according to which we represent the desired function as a product [17,18,19,20].
(9)$$G_{w}\left(r,\tau ;\xi \right)=R\left(r\right)T\left(\tau \right).$$

Substituting (9) into the homogeneous equation of motion (6) and separating the variables, we obtain the eigenvalue problem (λ is the separation constant).
(10)$$\Delta^{2}R\left(r\right)-\lambda R\left(r\right)=0;$$
(11)$${\left.R\left(r\right)\right|}_{r=1}=0,{\left.\;\left(\frac{d^{2}R\left(r\right)}{dr^{2}}+\frac{ℵ}{r}\frac{dR\left(r\right)}{dr}\right)\right|}_{r=1}=0.$$
(12)$$R\left(r\right)=O\left(1\right),\frac{dR\left(r\right)}{dr}=O\left(r\right),r\to 0.$$

1. Considering the case for λ>0, the solution of Equation (10) is sought in the form of eigenfunctions of the Laplace operator:(13)$$\Delta R\left(r\right)=\zeta R\left(r\right).$$

Substituting (13) into (10), we obtain the characteristic equation:(14)$$\zeta^{2}-\mu^{4}=0,\mu =\sqrt[{4}]{\lambda }.$$

Determining its roots $\zeta_{1,2}=\pm \mu^{2}$, we arrive at an equivalent set of equations:(15)$$\Delta R\left(r\right)+\mu^{2}R\left(r\right)=0,\;\Delta R\left(r\right)-\mu^{2}R\left(r\right)=0.$$

The general solution of Equation (10) has the following form: (16)$$R\left(r\right)=C_{1}J_{0}\left(\mu r\right)+D_{1}N_{0}\left(\mu r\right)+C_{2}I_{0}\left(\mu r\right)+D_{2}K_{0}\left(\mu r\right)..$$
where $J_{0}\left(z\right)$ and $N_{0}\left(z\right)$ are the Bessel and Neumann functions, respectively, and $I_{0}\left(z\right)$ and $K_{0}\left(z\right)$ are the modified Bessel and Macdonald function, respectively. Since the functions $N_{0}\left(\mu r\right)$ and $K_{0}\left(\mu r\right)$ are unlimited at r→0, then we need to establish that $D_{1}=D_{2}=0$, so: (17)$$R\left(r\right)=C_{1}J_{0}\left(\mu r\right)+C_{2}I_{0}\left(\mu r\right).$$

Now substituting (17) into boundary conditions (11), we arrive at a system of linear algebraic equations with respect to $C_{1}$ and $C_{2}$:(18)$$C_{1}J_{0}\left(\mu \right)+C_{2}I_{0}\left(\mu \right)=0.\;C_{1}\left[\frac{1-ℵ}{\mu }J_{1}\left(\mu \right)-J_{0}\left(\mu \right)\right]+C_{2}\left[\frac{ℵ-1}{\mu }I_{1}\left(\mu \right)+I_{0}\left(\mu \right)\right]=0.$$

A necessary and sufficient condition for the existence of a nontrivial solution of this system is the equality to zero of the determinant of its matrix [21,22,23]. Then, we obtain the following: (19)$$\frac{ℵ-1}{\mu }\left[J_{0}\left(\mu \right)I_{1}\left(\mu \right)+J_{1}\left(\mu \right)I_{0}\left(\mu \right)\right]+2J_{0}\left(\mu \right)I_{0}\left(\mu \right)=0.$$

This transcendental equation defines a countable set of non-negative roots $\mu_{n}$ [6,24,25]. Then, using the proper form of $R_{n}\left(r\right)$ from Equation (18), we attain: (20)$$R_{n}\left(r\right)=J_{0}\left(\mu_{n}r\right)-\frac{J_{0}\left(\mu_{n}\right)}{I_{0}\left(\mu_{n}\right)}I_{0}\left(\mu_{n}r\right).$$

2. For the case λ=0, the general solution of Equation (10) is found by its successive integration:(21)$$R_{0}\left(r\right)=C_{1}+C_{2}r^{2}+C_{3}\ln r+C_{4}r^{2}\ln r.$$

By virtue of conditions (12), we establish that $C_{3}=C_{4}=0$, so:(22)$$R_{0}\left(r\right)=C_{1}+C_{2}r^{2}.$$

A condition for the existence of an eigenvalue $\lambda_{0}=0$ is possible only under specific boundary conditions, and we will not consider it.

After defining our own forms of $R_{n}\left(r\right)$ and our own values of $\mu_{n}$, we represent the desired influence function in the form of a Fourier series:(23)$$G_{w}\left(r,\tau ;\xi \right)=\sum_{n}^{}G_{n}\left(\tau ;\xi \right)R_{n}\left(r\right).$$

The coefficients of the series $G_{n}\left(\tau ;\xi \right)$ are determined by known relations [14,26,27,28,29,30]. Then, the influence functions for other kinematic and static parameters of the plate will have the following form:(24)$$\begin{array}{r} G_{\vartheta }(r,\tau ;\xi )=\sum_{n}^{}G_{n}(\tau ;\xi )R_{\vartheta n}\left(r\right),\\ G_{M}(r,\tau ;\xi )=\sum_{n}^{}G_{n}(\tau ;\xi )R_{Mn}\left(r\right),\\ G_{Q}(r,\tau ;\xi )=\sum_{n}^{}G_{n}(\tau ;\xi )R_{Qn}\left(r\right), \end{array}$$
where
(25)$$\begin{array}{r} R_{\vartheta n}(r)=-\frac{dR_{n}(r)}{dr},R_{Mn}(r)=-\frac{d^{2}R_{n}(r)}{dr^{2}}-\frac{ℵ}{r}\frac{dR_{n}(r)}{dr},\\ R_{Qn}(r)=-\frac{d^{3}R_{n}(r)}{dr^{3}}-\frac{d^{2}R_{n}(r)}{rdr^{2}}+\frac{dR_{n}(r)}{r^{2}dr}. \end{array}$$

The time derivative of the influence functions is defined in the usual way: (26)$$\begin{array}{r} \frac{\partial G_{w}\left(r,\tau ;\xi \right)}{\partial \tau }=\sum_{n}\frac{\partial G_{n}\left(\tau ;\xi \right)}{\partial \tau }R_{n}\left(r\right),\\ \frac{\partial G_{\vartheta }\left(r,\tau ;\xi \right)}{\partial \tau }=\sum_{n}\frac{\partial G_{n}\left(\tau ;\xi \right)}{\partial \tau }R_{\vartheta n}\left(r\right),\\ \frac{\partial G_{M}\left(r,\tau ;\xi \right)}{\partial \tau }=\sum_{n}\frac{\partial G_{n}\left(\tau ;\xi \right)}{\partial \tau }R_{Mn}\left(r\right),\\ \frac{\partial G_{Q}\left(r,\tau ;\xi \right)}{\partial \tau }=\sum_{n}\frac{\partial G_{n}\left(\tau ;\xi \right)}{\partial \tau }R_{Qn}\left(r\right). \end{array}$$

As an example, we consider a hinged round plate under a sudden application of external pressure created by a water droplet impact. External pressure is determined from experimental studies. Depending on the jet velocity and the distance from the nozzle (Figure 2), the external pressure $p\left(\tau ,r\right)$ is determined. To illustrate the proposed method, we will assume that the droplet impact load is suddenly applied, then we can establish that $p\left(\tau ,r\right)=p_{0}H(\tau )$. The initial conditions are homogeneous. The plate material has average characteristics, reflecting in the first approximation the composite material of the plate $\left(ℵ=0.764\;\frac{h}{R}=0.01,\;p_{0}=1\right)$. Then, for given homogeneous initial conditions $\varphi \left(r\right)=\psi \left(r\right)=0$ and the applied water droplet impact, the deflection of the plate is determined by the ratio following from Equation (5): (27)$$w\left(\tau ,r\right)=p_{0}\int_{0}^{\tau }\int_{0}^{1}G_{w}\left(r,\tau ;\xi \right)d\xi .$$

At the same time, the function $G_{w}\left(r,\tau ;\xi \right)$ is determined from (23), then by calculating the double integral (27), we obtain the required deflection of the plate: (28)$$w\left(r,\tau \right)=p_{0}H(\tau )\sum_{n=1}^{\infty }\frac{q_{n}R_{n}(r)}{\mu_{n}^{4}\Delta }\left(1-\cos\omega_{n}\tau \right).$$

In this expression, $H\left(\tau \right)$ is the Heaviside function, $\omega_{n}=\frac{\mu_{n}^{2}}{\gamma }$, $\Delta =\frac{ℵ-1}{\mu }\left[J_{0}\left(\mu \right)I_{1}\left(\mu \right)+J_{1}\left(\mu \right)I_{0}\left(\mu \right)\right]+2J_{0}\left(\mu \right)I_{0}\left(\mu \right),$
$q_{n}=\frac{1}{\mu_{n}}\left[J_{0}\left(\mu_{n}r\right)-\frac{J_{0}\left(\mu_{n}\right)}{I_{0}\left(\mu_{n}\right)}I_{0}\left(\mu_{n}r\right)\right]$.

The bending moment is found by the formula $M=\frac{\partial^{2}w}{\partial r^{2}}+\frac{ℵ}{r}\frac{\partial w}{\partial r}$ using relation (25) by means of direct differentiation of the series (28): (29)$$M\left(r,\tau \right)=p_{0}H(\tau )\sum_{n=1}^{\infty }\frac{q_{n}R_{Mn}(r)}{\mu_{n}^{4}\Delta }\left(1-\cos\omega_{n}\tau \right).$$

Figure 5 and Figure 6 show the spatio-temporal and spatial dependencies for deflection $w\left(r,\tau \right)$, and in Figure 7 and Figure 8, there are similar dependencies for the moment $M\left(r,\tau \right)$. All calculations took into account five members of the series.

In this work, a single-jet impact platform is created based on a gas gun, which can generate stable high-velocity water jets. Next, tests are carried out for jet water droplet impact at various speeds for various types of obstacles. Based on the tests carried out, the external load is determined.

The results show that when exposed to a high-velocity jet impact, the surfaces of the tested structures mainly show damage in the form of subsurface damage. Observation of the stress wave propagation inside the sample showed the presence of shear waves.

Further, on the basis of the research carried out, mathematical modeling of the processes of a water droplet impact with an obstacle is carried out. To achieve this, dynamic equations of the linear theory of elasticity are solved for an annular plate for isotropic materials, in which the mechanical properties are close to the average characteristics of barriers made from composite materials.

## 5. Conclusions

The conclusions are as follows:Various forms of damage were experimentally established under the influence of a jet simulating a water droplet impact. Surface damage and crack laminations are shown;With increasing impact speed, the damage area of the two samples gradually increased, and peeling damage appeared. A composite barrier is more prone to spalling over a larger damage area at higher impact velocities than a monolithic plate;By observing the propagation of the stress wave and the behavior of the damage inside the samples, it was found that the form of damage to the barrier is subsurface delamination due to the predominance of the shear wave, which allows for modeling the water droplet impact load in the form of a suddenly applied pressure;At the found water droplet impact load, the stress–strain state of the barrier is definite, which is modeled by an isotropic round plate with averaged characteristics;The stress–strain state of the plate is modeled on the basis of the dynamic theory of elasticity using the influence functions for the kinematic and static parameters of the plate;As a result of solving the problem, the values of deflections and bending moments of the plate are obtained. The distribution of these parameters along the radius and over time is shown.

## Figures and Tables

**Figure 1 materials-15-07305-f001:**
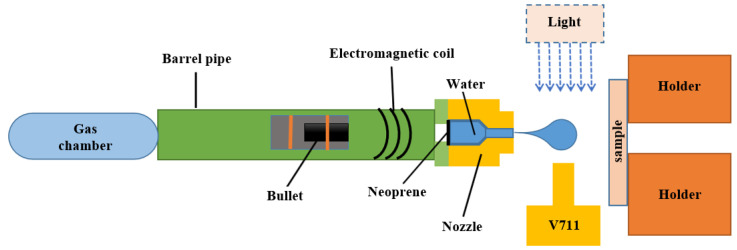
Platform for single jet impact testing.

**Figure 2 materials-15-07305-f002:**
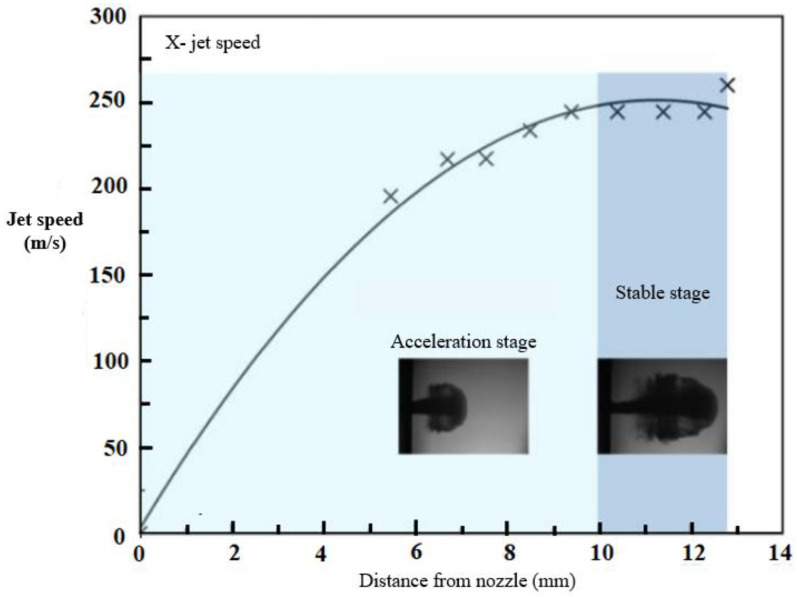
Changing the shape and speed of the jet with displacement.

**Figure 3 materials-15-07305-f003:**
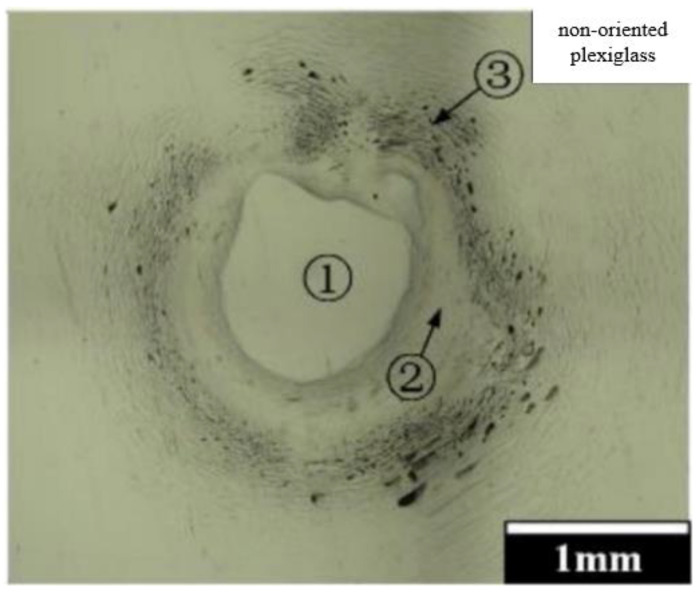
Microscopic images of the surface of two plexiglass samples after jet impact.

**Figure 4 materials-15-07305-f004:**
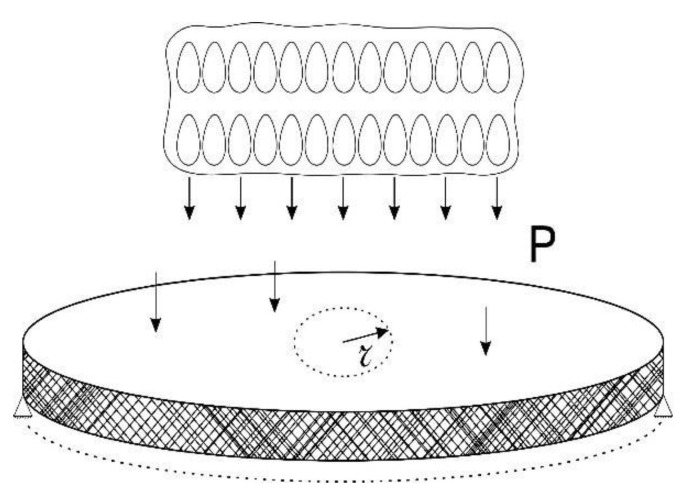
Drop-impact effect on the surface of the plate.

**Figure 5 materials-15-07305-f005:**
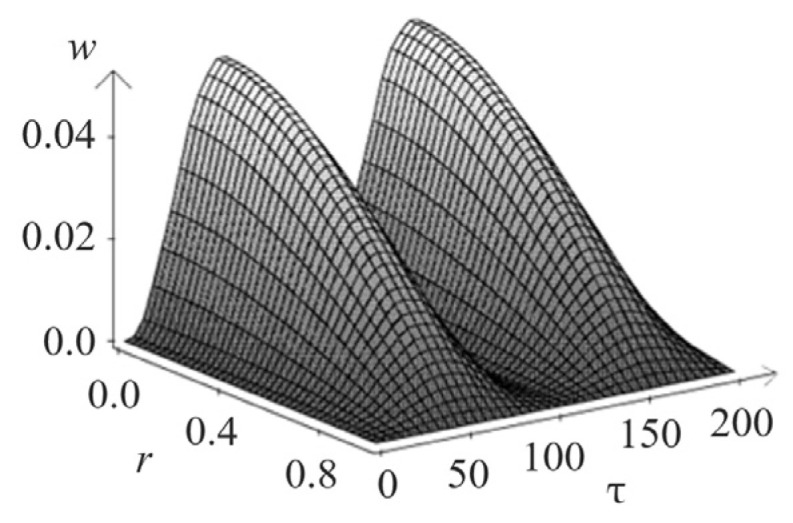
Spatio-temporal dependencies for deflection $w\left(r,\tau \right)$.

**Figure 6 materials-15-07305-f006:**
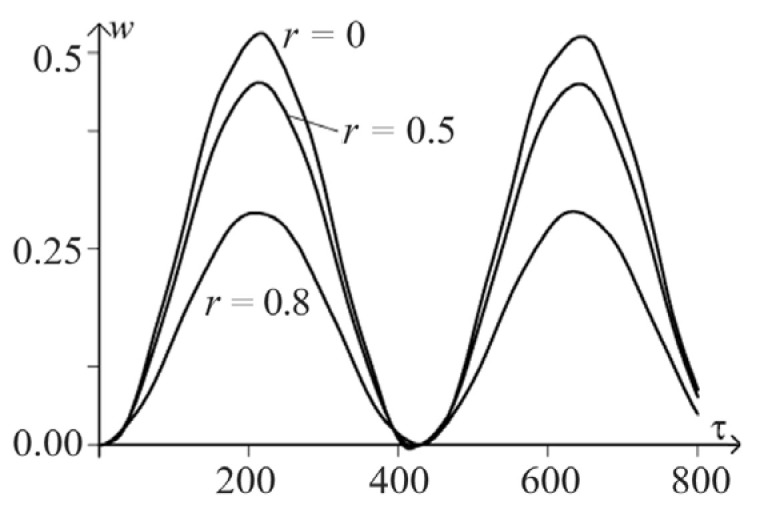
Temporal and spatial dependencies for deflection $w\left(r,\tau \right)$.

**Figure 7 materials-15-07305-f007:**
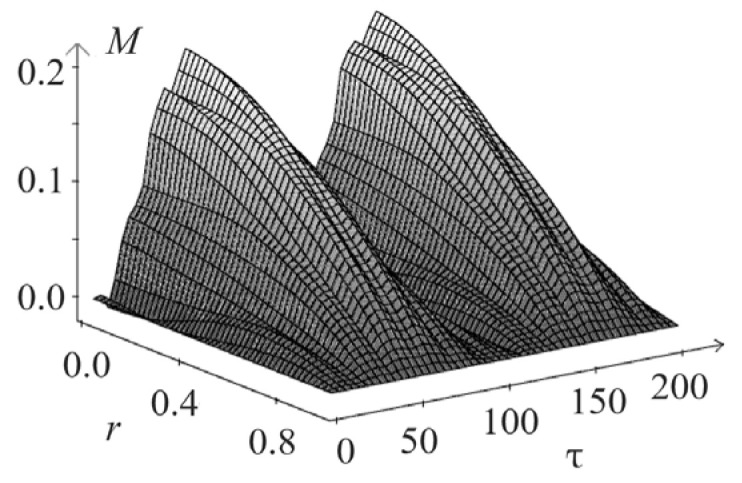
Spatio-temporal dependencies for deflection $M\left(r,\tau \right)$.

**Figure 8 materials-15-07305-f008:**
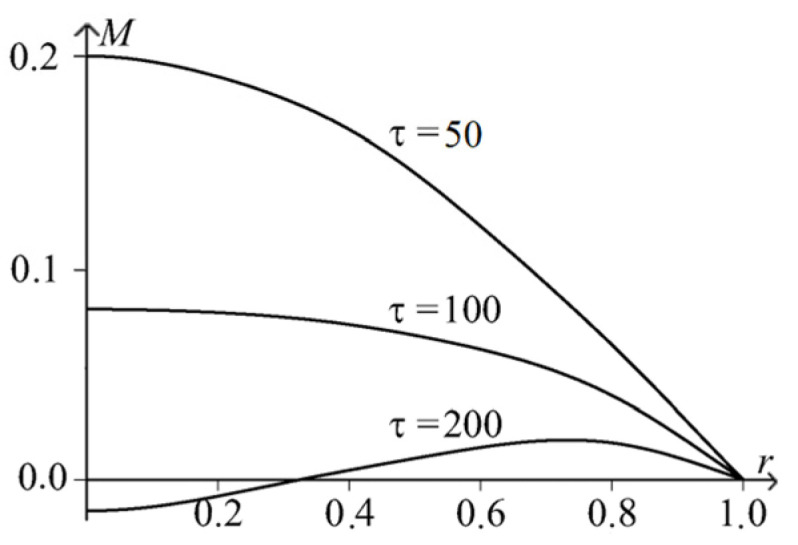
Temporal and spatial dependencies for deflection $M\left(r,\tau \right)$.

## Data Availability

Not applicable.

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
