# Peer review of "Theoretical and Experimental Studies of the Impact of High-Speed Raindrops on the Structural Elements of Modern Technology"

_materials, 2022, doi:10.3390/ma15207305_

Round 1

Reviewer 1 Report

Please see my comments in the following paragraphs. The topic is interesting and worthy of investigation. There are a few minor points that have to be addressed.

Comment (1): The significance and novelty of the research must be highlighted in the last paragraph of the introduction section.

Comment (2): Fig. 2: The authors should add enhanced-resolution images to show the accelerating and stable stages. The resolution of the present pictures is poor. Please add a scale bar to the photos.

Comment (3): In line 88, one of the dots must be removed.

Comment (4): In line 93, “The” must be removed from the beginning of the sentence.

Comment (5): Fig. 3: One of the sub-images is missed in the version of the paper sent out for peer review. Because the Fig is incomplete, I cannot comment on the corresponding discussion. I see Fig. 3a, not Fig. 3b.  

Comment (6): Please enhance the resolution of the images in Fig. 4 and clearly define the parameters shown in the photos.

Author Response

Comment (1): The significance and novelty of the research must be highlighted in the last paragraph of the introduction section.
Answer (1): We added in the Introduction the following text: “The non-stationary interaction of an inhomogeneous round plate with an external load simulating a high-velocity rain pressure is considered for the first time. A mathematical model is constructed, on the basis of which the kinematic and dynamic parameters of the system are obtained. Based on the obtained results, it is possible to predict the stress-strain state of aircraft structural elements during the flight through the rain zone at a high speed”.

Comment (2): Figure 2: The authors should add enhanced-resolution images to show the accelerating and stable stages. The resolution of the present pictures is poor. Please add a scale bar to the photos.
Answer (2): Figure 2 now has a higher resolution.

Comment (3): In line 88, one of the dots must be removed.
Answer (3): In line 88, one of the dots has been removed.

Comment (4): In line 93, “The” must be removed from the beginning of the sentence.
Answer (4): In line 93, “The” has been removed from the beginning of the sentence.

Comment (5): Fig. 3: One of the sub-images is missed in the version of the paper sent out for peer review. Because the Fig is incomplete, I cannot comment on the corresponding discussion. I see Fig. 3a, not Fig. 3b.
Answer (5): There are no sub-images in Figure 3 now. There is only one figure.

Comment (6): Please enhance the resolution of the images in Fig. 4 and clearly define the parameters shown in the photos.
Answer (6): The resolution of Figure 4 has been increased.

Reviewer 2 Report

This manuscript focused on the theoretical and experimental studies of the impact of high-speed raindrops on the structural elements of modern technology. The external load is determined on the basis of experimental studies using a single-jet impact platform and the tests are carried out for jet water droplet impact at various speeds for various types of obstacles. Finally, mathematical modeling of the processes of a water droplet impact with an obstacle is established. The results of this manuscript are reasonably interesting and novelty is easily identified. In my viewpoint, I recommend this manuscript to be accepted for publication after a minor revision. Some suggestions for the authors to improve the manuscript:

(1) Figure 3(b) needs to be added and the corresponding analysis should be supplemented.

(2) It is recommended that the Results and Discussion sections be combined for an in-depth analysis.

Author Response

Comment (1): Figure 3(b) needs to be added and the corresponding analysis should be supplemented.
Answer (1): There are no sub-images in Figure 3 now. There is only one figure.

Comment (2): It is recommended that the Results and Discussion sections be combined for an in-depth analysis.
Answer (2): Results and Discussion sections are combined into one.

Reviewer 3 Report

The study reported contains significant mathematical model merits, as it develops a model to determine deflectiotns caused by water droplets. 

How is the deformation of the water droplet considered here?, or are the auhors considering solid droplets only?

On the oher hand the article lacks ecessary experiemental data that confirm the developed model. The conclusions raised, as written, seem to suggest so, but there is not any result focussing on that.

Author Response

Comment (1): How is the deformation of the water droplet considered here?, or are the auhors considering solid droplets only?
Answer (1): In this paper, liquid drops are assumed to be spherical in shape and considered to be absolutely solid. We added to article the following text: «The raindrops are assumed to be spherical and uniform in size and do not deform».

Comment (2): On the other hand the article lacks necessary experimental data that confirm the developed model. The conclusions raised, as written, seem to suggest so, but there is not any result focusing on that.
Answer (2): An explanation of the recommendations made by the reviewer has been added to the text of the article. We added to article the information that «The presented results of experimental studies allow concluding that at a sufficient distance from the nozzle, the jet velocity has a steady character (Figure 2). Based on this, in the first approximation, it is possible to consider the external load acting on the barrier as suddenly applied, which makes it possible to use a mathematical model of the interaction of round plates under non-stationary impact».